# The EV71 2A protease occupies the central cleft of SETD3 and disrupts SETD3-actin interaction

Xiaopan Gao [1,4], Bei Wang[1,4], Kaixiang Zhu[1,4], Linyue Wang[1,4], Bo Qin[1], Kun Shang[2,3], Wei Ding [2] ✉, Jianwei Wang [1] ✉ & Sheng Cui [1] ✉

SETD3 is an essential host factor for the replication of a variety of enteroviruses that specifically interacts with viral protease 2A. However, the interaction between SETD3 and the 2A protease has not been fully characterized. Here, we use X-ray crystallography and cryo-electron microscopy to determine the structures of SETD3 complexed with the 2A protease of EV71 to 3.5 Å and 3.1 Å resolution, respectively. We find that the 2A protease occupies the V-shaped central cleft of SETD3 through two discrete sites. The relative positions of the two proteins vary in the crystal and cryo-EM structures, showing dynamic binding. A biolayer interferometry assay shows that the EV71 2A protease outcompetes actin for SETD3 binding. We identify key 2A residues involved in SETD3 binding and demonstrate that 2A's ability to bind SETD3 correlates with EV71 production in cells. Coimmunoprecipitation experiments in EV71 infected and 2A expressing cells indicate that 2A interferes with the SETD3-actin complex, and the disruption of this complex reduces enterovirus replication. Together, these results reveal the molecular mechanism underlying the interplay between SETD3, actin, and viral 2A during virus replication.

Enteroviruses (EVs) belonging to the Picornaviridae family include many severe human pathogens[1]. For example, a group of human EVs including enterovirus 71 (EV71) and coxsackievirus A16 (CA16) are the causative agents of hand, foot, and mouth disease (HFMD). Although HFMD is generally mild, EV71 infection damages the central nervous system (CNS), and in severe cases can cause death[2]. Poliovirus (PV) is another CNS-invading EV that can paralyze a patient in hours[3]. Despite the launch of the Global Polio Eradication Initiative in 1988, polio eradication remains unaccomplished[4]. Especially, the recent return of PV to several polio-free countries is alarming[5]. Currently, there are no approved drugs for EV infections.

Although EVs encapsulate a plus single-stranded RNA genome of only ~7 kb, the genome encodes two cysteine proteases, 2A and 3C, highlighting their importance during infection[6]. While 3C mediates most polyprotein processing[7–9], 2A only mediates the cleavage between VP1-2A, and it is more involved in subverting host cell functions, including host protein synthesis shutdown, nucleocytoplasmic transportation inhibition, and innate immunity evasion[10–13]. Structural characterization revealed that EV 2A adopts a chymotrypsin-like fold comprising two domains (I and II), between which lies the protease active site[14,15]. Additionally, 2A contains a zinc finger in domain II, the ligands for tetrahedral coordination of the zinc are donated by the C56, C58, C116 and H118[14], implying its involvement in protein-protein interactions[16].

Recently, SET domain-containing protein 3 (SETD3) was identified as a host factor crucial for EV infection and pathogenesis[17]. SETD3 interacts with 2A from different EVs, including EV71, coxsackievirus B3 (CVB3), and PV. The SETD3-2A interaction was found to be essential for

[1]NHC Key Laboratory of Systems Biology of Pathogens, National Institute of Pathogen Biology, Chinese Academy of Medical Sciences and Peking Union Medical College, Beijing 100730, China. [2]Beijing National Laboratory for Condensed Matter Physics, Institute of Physics, Chinese Academy of Sciences, Beijing 100190, China. [3]Medical School, Yan'an University, Yan'an, China. [4]These authors contributed equally: Xiaopan Gao, Bei Wang, Kaixiang Zhu, Linyue Wang. ✉e-mail: dingwei@iphy.ac.cn; wangjw28@163.com; cui.sheng@ipb.pumc.edu.cn

viral RNA replication. SETD3 functions as an actin-specific histidine N-methyltransferase, and comprises an N-terminal SET domain and a C-terminal Rubisco large subunit methyltransferase (LSMT) like domain. The N- and C-terminal domains of SETD3 form a V-shaped central cleft, within which the actin substrate is recognized and methylated at residue H73[18,19]. Several structures of SETD3 complexed with an H73-containing actin peptide revealed structural details for its sequence-specific peptide recognition, as well as the methylation mechanism at H73[19–21].

Although structures of SETD3 in complex with actin peptide have been determined, the mode of the SETD3-actin interaction involving intact proteins remains unclear. By comparing an actin monomer with SETD3, it is obvious that the SETD3 central cleft is too narrow for fitting a whole actin molecule, and a dramatic conformational change of actin is required for it to gain access to the SETD3 active site. By contrast, the size of 2A suggests it might fit well into the SETD3 central cleft (Supplementary Fig. 1), which implies that 2A may disrupt the SETD3-actin complex through competitive binding.

Here, we determine the X-ray crystallography and cryo-electron microscopy structures of SETD3 complexed with the 2A protease of EV71, respectively. The 2A protease occupies the V-shaped central cleft of SETD3 through two distinct sites. A biolayer interferometry assay reveals that the EV71 2A protease competitively binds to SETD3 over actin. EV71-infected and 2A-expressing cells suggest that 2A disrupts the SETD3-actin complex, leading to reduced enterovirus replication. Our findings unveil the molecular mechanism governing the interaction among SETD3, actin, and viral 2A during virus replication.

## Results and discussion

### Assembly of the SETD3/EV71 2A complex

To gain experimental evidence for SETD3-2A interaction, we first investigated the interaction between SETD3 and 2A by coexpression and copurification from insect cells. Because wild-type 2A expression resulted in shutdown of host protein translation, and hinders protein production, we produced protease-inactive mutants EV71 2A C110A and CVB3 2A C107A for most biophysical and structural characterizations. We coexpressed full-length (FL) SETD3 protein or SETD3 variants lacking ~100 C-terminal amino acids (residues 1–503) with EV71 2A in insect cells (Fig. 1a, b, Supplementary Fig. 2a), in which no affinity tag was fused to 2A, and an N-terminal 6×His-tag was engineered to SETD3 or its variants. Both SETD3 FL and SETD3 1–503 coeluted with EV71 2A from Ni-NTA resin, implying the formation of stable SETD3-2A complexes (Fig. 1 a, b). To rule out the possibility of other molecules being involved in the SETD-2A complexes, we separately produced SETD3(1–498) and EV71 2A using *Escherichia coli* (Supplementary Fig. 2b), then mixed the two proteins in vitro. We preincubated SETD3 1–498 with EV71 2A at a molar ratio of 1:3 before loading onto a size-exclusion column. As shown in Fig. 1c, d, SETD3 and 2A coeluted as a stable complex.

To confirm the stoichiometry of the SETD3-EV71 2A complex, we conducted analytical ultracentrifugation (AUC) experiments (Fig. 1 e). The molecular mass of EV71 2A alone determined by AUC is ~17 kDa; the molecular mass of SETD3 (1-498) determined by AUC is ~54 kDa, close to their theoretical molecular mass, 16 kDa for 2A and 57 kDa for SETD3. The calculated molecular mass of the SETD3-EV71 2A complex is ~66 kDa by AUC, it is very close to its theoretical molecular mass, 73 kDa. In summary, AUC results support 1:1 stoichiometry of the SETD3-EV71 2A complex (Fig. 1e).

Because SETD3 FL was unstable after purification (Supplementary Fig. 2a), and lack of the C-terminal portion did not affect the SETD3-2A interaction (Fig. 1b, c), we chose the truncated SETD3 for further structural investigations. Specifically, SETD3 1–503 coexpressed with EV71 2A in insect cells was suitable for cryo-electron microscopy (EM) studies, and separately produced SETD3 1–498 and EV71 2A from *E. coli* were suitable for in vitro complex assembly and crystallization.

### SETD3 interacts with various EV 2A proteins

It was reported that SETD3 can interact with various enterovirus 2A proteins[17]. To identify the molecular determinant governing the SETD3-2A interactions, we assessed the binding affinity of SETD3 for various EV 2A proteinases using biolayer interferometry (BLI). We biotinylated SETD3 for loading on the streptavidin biosensors; the SETD3 loaded biosensors were then analyzed for binding with 2A protein in solution; the concentration series were 15.6-250 nM. The calculated dissociation constant ($K_D$) for SETD3-EV71 2A and SETD3-CVB3 2A interactions were 22.7 nM and 11.8 nM, respectively (Fig. 1f, g), indicating the SETD3-CVB3 2A interaction is merely ~ 2 folds higher than that of SETD3-EV71 2A. In order to investigate the role of S-adenosyl-L-homocysteine (SAH) in SETD3-2A binding, we added different amount of SAH in binding buffer and compared the SETD3-EV71 2A binding kinetics (Fig. 1h). The presence of SAH had negligible effects on binding.

These results imply that the mode of interaction between SETD3 and various 2A proteins is similar. Although the amino acid sequence identity shared among different viral 2A proteins was 35–71%, variation in different 2A sequences might not contribute to their affinities for SETD3. Our following structural investigations reveal that SETD3 binds a highly conserved region on 2A. Considering availability of protein constructs, virus strains, antibodies and infection clones, we chose EV71 as the model organism and focused on SETD3-EV71 2A interactions in current study.

### Integrative structure determination SETD3 complexed with EV71 2A

To reveal the structural basis for the SETD3-2A interaction, we carried out crystallization for in vitro-assembled SETD3-2A complexes, and only SETD3 1–498 mixed with EV71 2A yielded protein crystals, and we tested ~1,000 crystals for X-ray diffraction. Whereas most diffracted X-rays to more than 6 Å, the best crystal diffracted to 3.5 Å resolution. It belonged to the $P2_1$ space group, and the structure was solved by molecular replacement using PDB 3W95and 6MBKas templates. There were four copies of the SETD3-2A complex in the asymmetric unit. However, only chains B and F in one copy had relatively clear electron density, hence they were used for structural analyses (Fig. 2a, left). We clearly identified electron density for the zinc chelated by the zinc finger of 2A (Fig. 2a) even though zinc ions were not supplemented during purification. We validated the presence of zinc in the structure by calculating polder maps at the 2A zinc finger, which revealed positive difference densities for zinc ion. Statistics for data collection and refinement are summarized in Supplementary Table 1.

The poor resolution of our X-ray diffraction data motivated us to switch to a single-particle cryo-EM approach for structural investigation. The coexpressed SETD3 1–503-EV71 2A complex from sf21 cells was suitable for sample vitrification and data collection. Although the molecular mass of this complex was only 73 kDa, we obtained a 3.1 Å cryo-EM density map providing further structural details (Fig. 2b). We could locate residues 21–501 of SETD3 and residues 6–146 of EV71 2A in the EM density (Fig. 2b, Supplementary Fig. 3&4, and Supplementary Table 2). While we independently completed our structural characterization of SETD3-EV71 2A complex, a 3.5 Å cryo-EM structure of SETD3-CVB3 2A complex was reported, which shows a similar mode of interaction between two proteins[22]. Superimposing our SETD3-EV71 2A structure with the CVB3 2A-SETD3 structure (PDB: 7LMS) gave an overall rmsd values of 1.44 Å and aligned 607 residues. As shown in Fig. 2d, the SETD3-2A interaction mode appears to be highly similar (Fig. 2d). We further compared residues at the SETD3-2A interface (Supplementary Fig. 5), demonstrating that residues of two viral 2A protease involved in the interaction with SETD3 are highly conserved.

Both our crystal and EM structures reveal that EV71 2A occupies the V-shaped central cleft of SETD3 formed between the SET and LSMT-like domains (Fig. 2a, b). Whereas domain II of 2A participates in

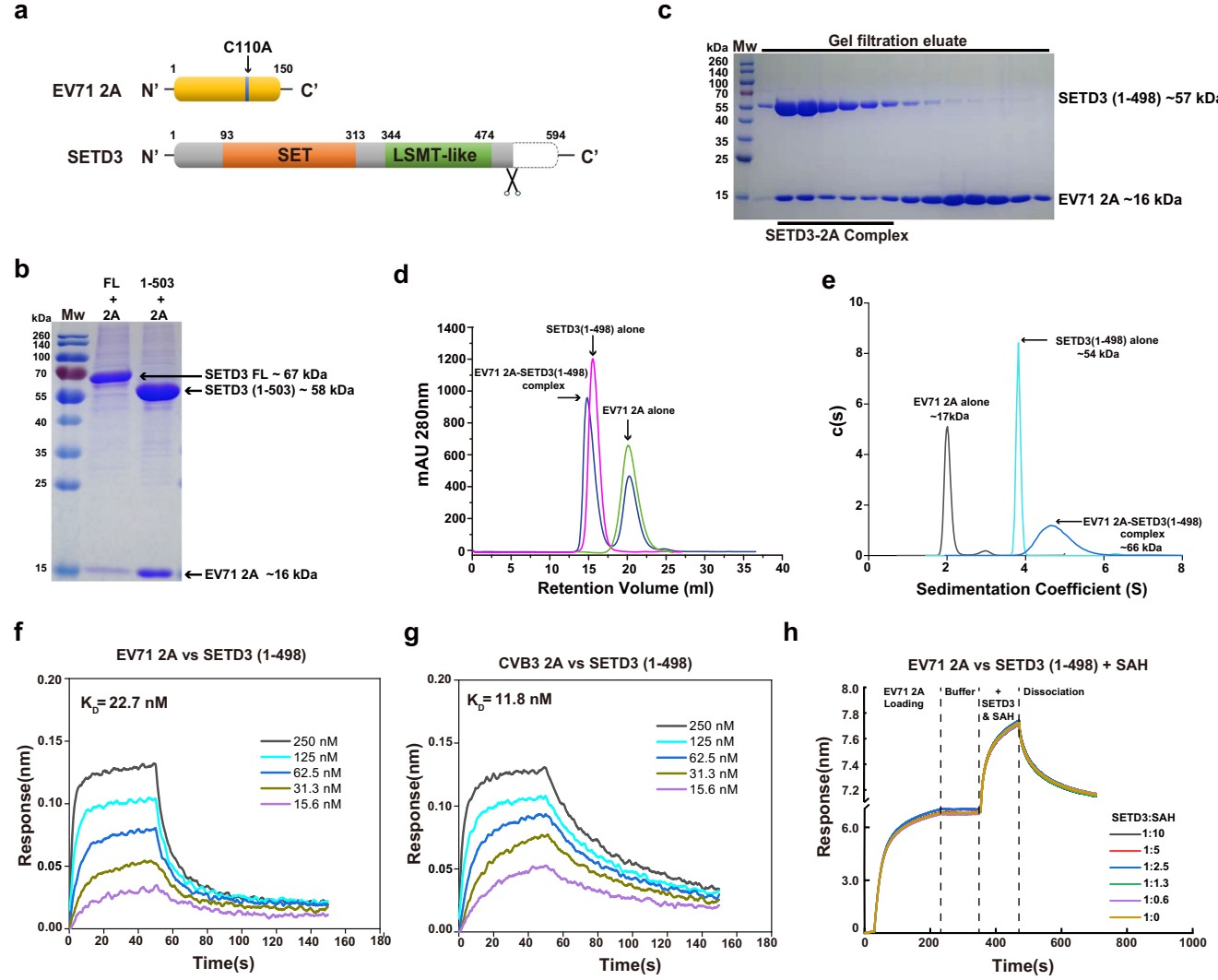

**Fig. 1 | SETD3 and EV71 2A Form a Stable Protein Complex In Vitro. a** Diagrams of protein constructs used for assembly of the SETD3-EV71 2A complex in vitro. Mutation C110A was introduced to EV71 2A during expression in *E. coli*. SETD3 1–498 was expressed in *E. coli*, and SETD3 1–503 was expressed in insect cells. **b** Coexpression and copurification of SETD3 and EV71 2A. Full-length SETD3 or SETD3 1–503 was coexpressed with EV71 2A, the complexes were purified using Ni-NTA resin, and elution was analyzed by SDS-PAGE. Data are representative of three independent experiments. Source data are provided as a Source Data file. **c** In vitro assembly of the SETD3-EV71 2A complex. Separately produced SETD3 1–498 and EV71 2A were mixed in vitro, and the mixture was loaded onto a Superdex 200 column. Eluate containing the protein complex was analyzed by SDS-PAGE. Data are representative of three independent experiments. Source data are provided as a Source Data file. **d** Size-exclusion chromatographic analysis of the SETD3-EV71 2A complex assembly using a Superdex 200 column. Source data are provided as a Source Data file. **e** Sedimentation analyses of the SETD3-EV71 2A complex assembly; the calculated molecular weights are indicated. Source data are provided as a Source Data file. **f** Bio-layer interferometry (BLI) analysis of SETD3 1–498 (immobilized on SA biosensor) for binding with EV71 2A; 2A concentration series and calculated dissociation constant ($K_D$) of binding are indicated. **g** BLI experiments of SETD3 1–498 for binding CVB3 2A; 2A concentration series and calculated dissociation constant ($K_D$) of binding are indicated. **h** BLI experiments of SETD3 1–498 for binding with EV71 2A in the presence of SAH; molar ratio of SETD3:SAH is indicated.

binding to the V-shaped cleft, no contacts were found between 2A domain I and SETD3 (Fig. 3a). There are two discrete contacts between SETD3 and EV71 2A domain II, denoted Site-1 and Site-2. The interfacial areas at Site-1 and Site-2 were calculated to be 728.6 Å² and 240.8 Å², respectively.

At Site-1, the three-stranded antiparallel β-plane (β-aII, β-bII1, and β-dII) on one side of 2A domain II is packed against a hydrophobic groove near the active site of the SET domain of SETD3 (Fig. 3b, left). In detail, interaction at Site-1 is centered by 2A residues V62, H71, and P73 that press against the groove formed by SETD3 residues N256, Q257, P259, T286, and Y288. Additionally, polar contacts including salt bridges and hydrogen bonds strengthen the SETD3-2A interaction; there are three hydrogen bonds (EV71 2A H71-SETD3 Y288, EV71 2A R115-SETD3 S264, and EV71 2A Q117-SETD3 Q257) and one salt bridge

(EV71 2A K70-SETD3 E296; Fig. 3b, left), and these residues are relatively conserved in different EVs (Supplementary Fig. 6). Residues R115 and Q117 belong to the EV71 2A zinc finger, underscoring its role in protein-protein interactions. It is worth noting that Site-1 overlaps the binding pocket for the actin peptide (Fig. 3c), suggesting EV 2A probably interferes with actin binding by occluding the SETD3 actin binding site[19].

In Site-2, EV71 2A interacts through the LSMT-like domain of SETD3 with the other side of domain II (Fig. 3a, b, right). SETD3 residue F409 extends its phenylalanine side chain against a hydrophobic patch on 2A domain II constituted by L81, Y83, and Y94. Additionally, SETD3 Q380 forms a hydrogen bond with EV71 2A K78. Together, our integrative structural characterizations revealed molecular determinants underlying the SETD3-2A interaction.

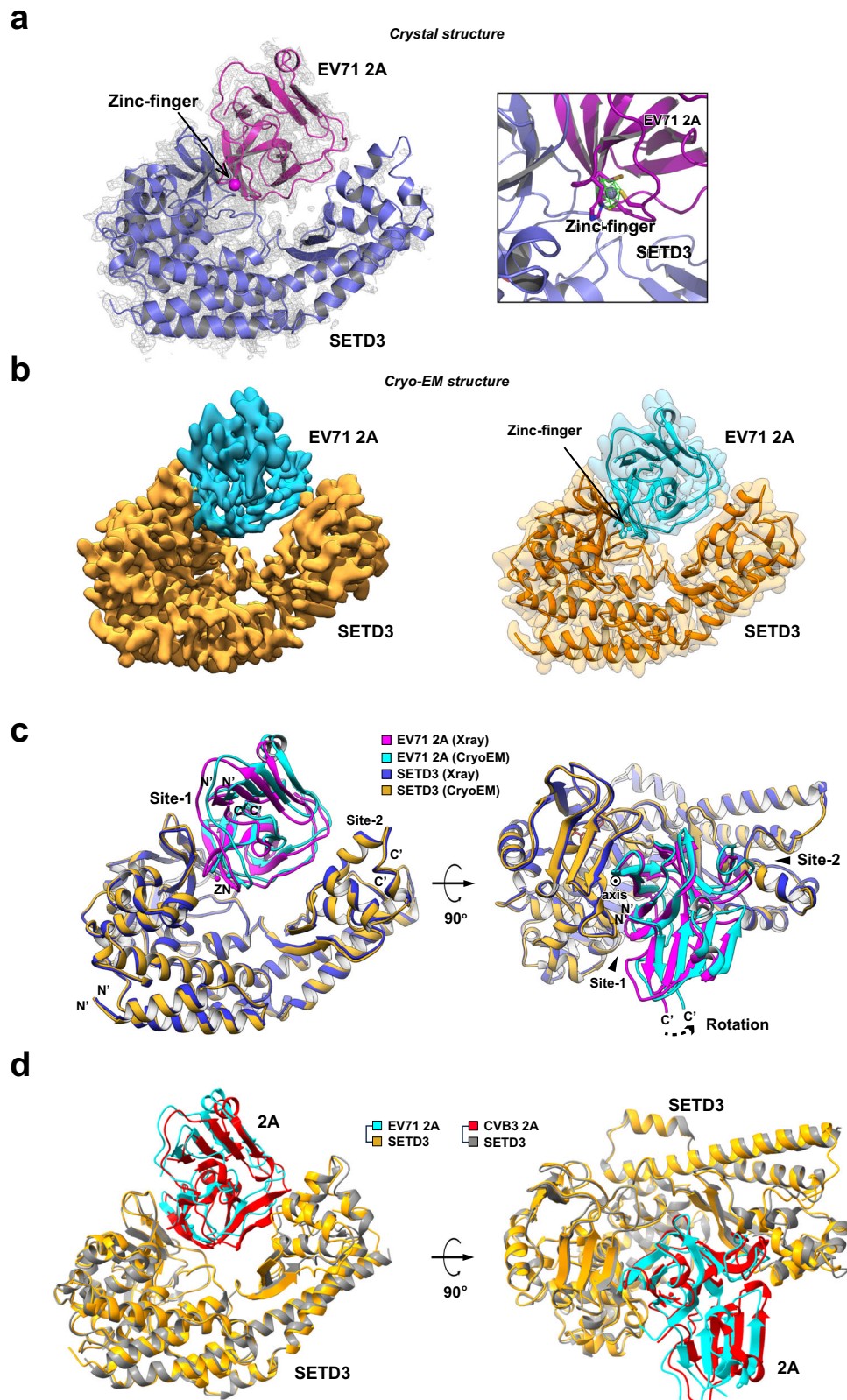

Comparing the two independently determined SETD3-2A complex structures showed that while the conformation of each component was highly similar (RMSD between the crystal and EM structures of EV71 2A was 0.49 Å, and that between SETD3 structures was 0.64 Å), their relative positions were slightly shifted, which contributes to the increased overall RMSD to 1.20 Å. We then aligned the crystal and EM structures with respect to SETD3, and identified a rotation between SETD3 and 2A (Fig. 2c). The rotational axis runs along the β-bII1 strand of EV71 2A, where 2A engages in an intermolecular antiparallel β-strand interaction with the SETD3 SET domain through two backbone hydrogen bonds (Fig. 3b, d). Structural variation between the crystal and EM structures reflects a degree of flexibility in the SETD3-2A interaction.

While domain II of EV71 2A mediates the binding with SETD3, the protease active site located at the gap between domain I and domain II

**Fig. 2 | Determination of the SETD3-EV71 2A complex structure by integrative approaches. a** Left, crystal structure (ribbon model) of SETD3 1–498 in complex with EV71 2A determined to 3.5 Å resolution; a final 2Fo-Fc electron density map (gray mesh, contour level 1.2σ) is superimposed on the model. The bound zinc ion is indicated with arrows. Right, magnified view of the EV71 2A zinc finger; a polder omit map for zinc contoured at ±3σ (magenta mesh) is superimposed on the model. **b** Left, cryo-EM density map (determined to 3.1 Å) for SETD3 1–503 in complex with EV71 2A; the structure is in the same orientation as in (**a**). The map for SETD3 is colored orange, and the map for EV71 2A is colored cyan. Right, semitransparent cryo-EM density map for the SETD3-EV71 2A complex superimposed on the final

atomic model of the complex (ribbon model with the same color scheme). **c** Final atomic models of the SETD3-EV71 2A complex derived from crystallography and cryo-EM are superimposed, with the SETD3 structure as the reference. While SETD3 and EV71 2A are colored blue and magenta in the crystal structure, the two components are colored orange and cyan in the cryo-EM structure. Structure alignment reveals that EV71 2A undergoes a subtle rotation around an axis (indicated by a circle with a central dot) running along the β-bII1 strand of EV71 2A. **d** Structural superimposition of the SETD3-EV71 2A (colored gold and cyan) and the SETD3-CVB3 2A complexes (colored gray and red, PDB: 7LMS).

(Fig. 3c middle). By superimposing a peptide bound EV71 2A structure (PDB: 4FVD) to the SETD3-EV71 2A complex, we modeled the substrate peptide into the protease active site (Fig. 3c right insert), which clearly shows the substrate has free access to EV71 2A active site in the presence of SETD3.

## Competition between 2A and actin binding to SETD3 in vitro

To understand the mechanism for 2A-mediated inhibition of actin binding to SETD3, we aligned the structures of unliganded SETD3 (PDB: 3SMT), actin peptide-bound SETD3 (PDB: 6OX2), and 2A-bound SETD3 (this study, Fig. 3c). We then calculated interface area between SETD3 and actin peptide, ~1069 Å$^2$, which is slightly larger than that between SETD3 and EV71 2A (~969 Å$^2$, Site-1 plus Site-2 combined). However, it is insufficient to judge which interaction is stronger depending on interfacial area comparison. Especially, the interface area calculated from the actin peptide-bound SETD3 structure cannot fully represent interaction between SETD3 and a folded actin protein. Structural superimposition revealed that a β-strand of SETD3 (in site-1) undergoes remarkable conformational rearrangements upon binding to substrates, actin peptide or 2A (Fig. 3c); hence, we denoted it the recognition strand (R-strand). The R-strand recognizes its substrates by shifting toward them by ~8 Å (Fig. 3c left insert), implying an induced fit mechanism during substrate recognition. Because 2A induced similar R-strand rearrangement to that of actin peptide, 2A was probably recognized by SETD3 as a substrate mimic. As the Fig. 3d illustrated, the 2A-binding site in SETD3(cyan stick model) partially overlaps with the actin peptide residues 66-72 binding site (red stick model) in SETD3, both forming intermolecular β sheets with SETD3.Two hydrogen bonds are formed between the backbone amide and carbonyl oxygen of SETD3 Y288 with carbonyl and amide of 2A H71. In the actin-SETD3 complex, the SETD3 Y288 amide forms a hydrogen bond with the carbonyl of actin Y69, and a second backbone hydrogen bond is formed between the SETD3 T286 carbonyl and the amide in the actin peptide I71. It is evident that actin peptide and EV71 2A compete for the same binding site in SETD3.

The 2A-binding site in SETD3 overlaps with the actin peptide-binding site (Fig. 3c, d), indicating a possible competition between 2A and actin for binding SETD3. To test this hypothesis, we first investigated whether EV71 2A can affect the methyltransferase (MTase) activity of SETD3. We determined steady-state kinetics of SETD3 for the actin peptide (66-TLKYPIEH_GIVTNWD-80) as described previously (Supplementary Table 3)[19]. The calculated Km was $8.1 \pm 0.5$ μM and the turnover number kcat was $18.5 \pm 0.3$ h$^{-1}$ (Fig. 3e, Supplementary Table 4). When adding EV71 2A with twofolds and fivefolds excess over SETD3 in the reaction mixtures, the Km of the MTase reaction increased to $38.9 \pm 6.8$ μM and $44.9 \pm 8.8$ μM, respectively; and the kcat dropped to $13.5 \pm 1$ h$^{-1}$ and $9.6 \pm 0.8$ h$^{-1}$, respectively (Fig. 3e, Supplementary Table 4). These results demonstrate that EV71 2A impaired actin-peptide binding at the SETD3 active site thus inhibited the MTase activity, which supports our hypothesis.

Next, we investigated competitive binding of EV71 2A and actin for SETD3 using BLI experiments. To this end, we prepared intact actin instead of an actin peptide. Given actin self-polymerization may hamper SETD3 binding, we introduced two mutations (A204E/P243K)

to obtain monomeric actin as previously described[23], denoted AP-actin (Supplementary Fig. 7). AP-actin was kept depolymerized in G buffer prior to experiments. As shown in Fig. 3f, binding of EV71 2A with SETD3 (immobilized on streptavidin biosensors) was barely disrupted by subsequent loading of AP-actin. By contrast, preloading AP-actin to SETD3 had negligible effects on subsequent binding of EV71 2A (Fig. 3g). Combining our enzymatic and biophysical investigations, we provide evidence that 2A can disrupt the SETD3-actin interaction through competitive binding and that 2A and actin cannot simultaneously bind with SETD3.

## The site-1 is critical for SETD3 binding and viral replication

To verify our structural findings, we carried out systematic mutagenesis using BLI, and EV71 reverse genetic system. We expressed a selection of EV71 2A mutants harboring alanine substitutions at Site-1 based on the C110A template and measured their binding affinities for SETD3 using BLI (Fig. 4a, Supplementary Fig. 8). All EV71 2A mutants were confirmed to be expressed and folded correctly and finally purified by size exclusion chromatography, exhibiting similar elution profiles as EV71 2A C110A mutant. While most mutations at Site-1 undermined the binding affinity (Fig. 4a, Supplementary Fig. 8), H71A elicited negligible effects. H71 contributed to binding through backbone hydrogen bonds, thus alanine substitution did not impair the interaction (Supplementary Fig. 8c). Combining the mutation KHY(70-72)AAA led to greater loss of binding affinity than either single mutation (Fig. 4a, Supplementary Fig. 8). Of note, triple mutation KHY(70−72)AAA that reduced its binding to SETD3 also impaired its ability of inhibiting AP-actin binding to SETD3 and MTase activity of SETD3 (Fig. 4b, c).

Next, we tested the impact of a set of 2A mutations on EV71 propagation using an EV71 reverse genetic system. We prepared EV71 infectious clones bearing key 2A mutations and transfected them into BSR/T7 cells to rescue virus. The supernatants of transfected cells were collected for infecting RD cells, and virus production was assessed by measuring VP1 expression (Fig. 4d). In Site-1, none of the single mutations R69A, K70A, H71A, or Y72A was sufficient to halt EV71 production; nevertheless, triple mutation KHY(70−72)AAA was lethal for EV71. These results further indicate that the SETD3-2A interaction is essential for EV71 replication.

## The 2A-mediated disruption of SETD3-actin complex is required for virus replication

Although the protease activity of 2A is important for EV replication[24], the activity has been demonstrated to be unrelated to SETD3-2A interaction[17,25]. Moreover, the viral infection does not necessitate the methyltransferase activity of SETD3[17]. Presently, there is no evidence to explain the precise molecular mechanism of how the interplay between the EV71 2A protease and SETD3 operates during viral replication. Based on our structural findings that 2A partially occupies the actin-binding site in SETD3 (Fig. 3c) and the BLI results showing 2A can outcompete with actin protein for SETD3 binding, (Fig. 3f, g), we speculated that the 2A-mediated disruption of SETD3-actin interaction is a key step during virus replication. To test this hypothesis, we first employed coimmunoprecipitation (Co-IP) experiment to investigate

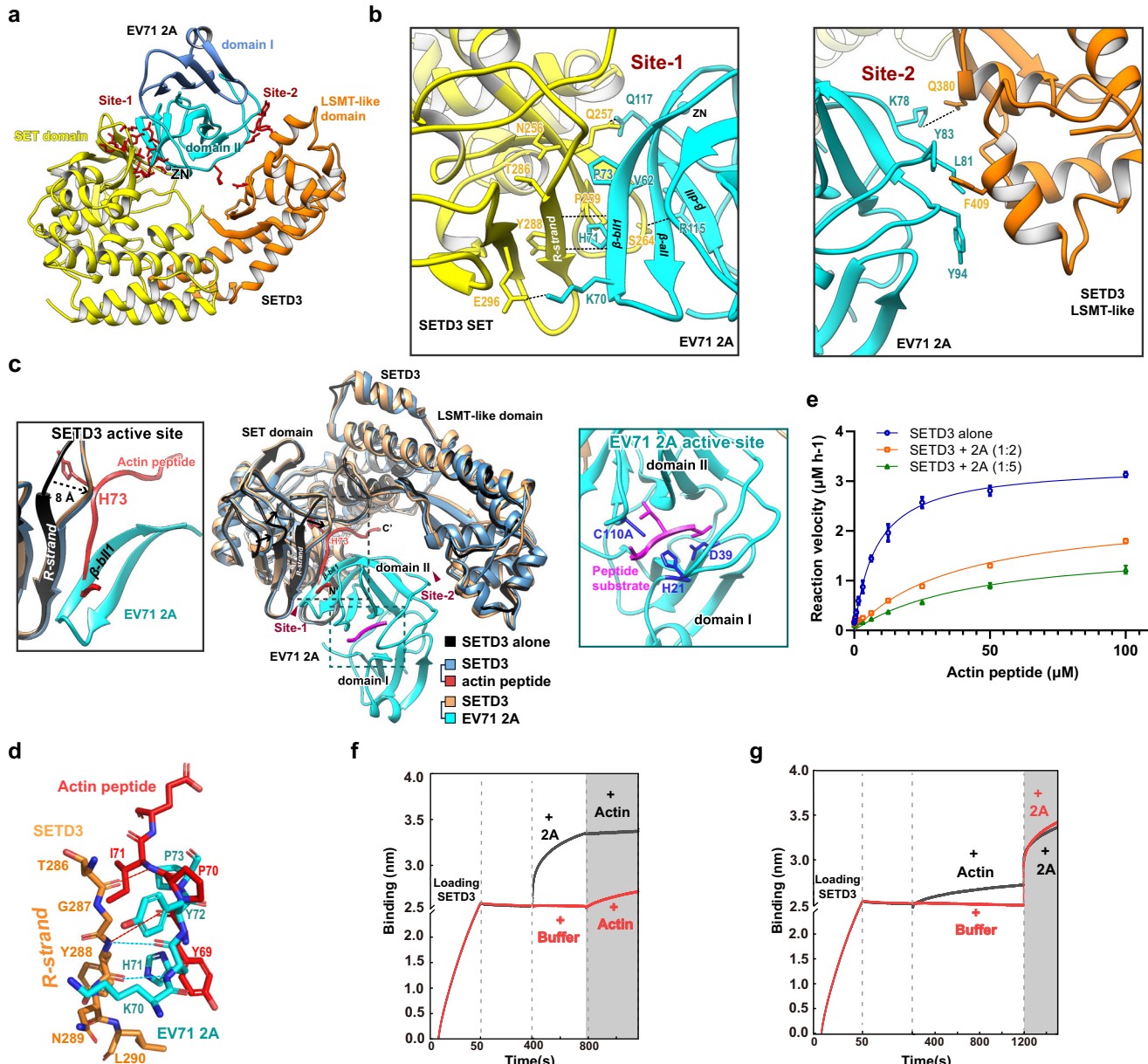

**Fig. 3 | Structural Basis for SETD3-EV71 2A Interactions. a** EV71 2A occupies the central cleft of SETD3, where it makes two discrete contacts with each domain of SETD3, denoted Site-1 and Site-2. **b** Left and right, details of intermolecular interactions at Site-1 and Site-2. Residues important for intermolecular interactions are shown in stick representation and labeled; hydrogen bonds and salt bridges are shown as dashed lines. **c** Structure superimposition of unliganded SETD3 (PDB: 3SMT), SETD3 complexed by actin peptide (PDB: 6OX2), and the cryo-EM structure of the SETD3-EV71 2A complex reported in this study. Binding by actin peptide or EV71 2A induces large conformational changes in the SETD3 SET domain, indicated by arrows. Left insert, magnified view of SETD3 active site, illustrating interaction details between SETD3-Actin peptide and between SETD3-EV71 2A. Right insert, magnified view of EV71 2A protease active site; substrate peptide (magenta) is modeled in the active site by superimposing an EV71 2A-peptide complex (PDB: 4FVD) with the EV71 2A chain in the complex. **d** The R-strand of SETD3 forms backbone hydrogen bonds with the β-bII1 strand of EV71 2A, where the β-bII1 strand would clash with the bound actin peptide. **e** The MTase activity of SETD3 is inhibited by EV71 2A. Blue curve, SETD3 alone; red curve, SETD3:EV71 2A molar ratio=1:2; green curve, SETD3:EV71 2A molar ratio = 1:5. Data are mean ± s.d. from three independent experiments (*n* = 3). Source data are provided as a Source Data file **f** BLI experiments illustrating preloading EV71 2A to SETD3 (immobilized on SA biosensor) was barely disrupted by subsequent loading of AP-actin. Black curve, SETD3 loaded biosensor was first exposed to EV71 2A and subsequently AP-actin; Red curve, SETD3 loaded biosensor was first exposed to G buffer and subsequently AP-actin. **g** BLI experiments illustrating preloading AP-actin to SETD3 (immobilized on SA biosensor) had negligible effect on subsequent binding of EV71 2A. Black curve, SETD3 loaded biosensor was first exposed to AP-actin and subsequently EV71 2A; Red curve, SETD3 loaded biosensor was first exposed to G buffer and subsequently EV71 2A.

the interplay among SETD3, actin, and 2A in EV71-infected cells (Fig. 4e). SETD3 and actin interacted directly with each other in mock-infected cells, whereas most actin molecules were stripped from SETD3 upon EV71 infection. This result confirms that the SETD3-actin complex was disrupted during viral infection (Fig. 4e). Similar results were observed in cells overexpressing 2A; 2A overexpression caused

the exclusion of actin from the SETD3-actin complex as compared to the untransfected cells (Fig. 4f). It should be noted that the total amount of actin in whole-cell lysates remained unchanged, thereby ruling out the possibility that the decrease of actin in Co-IP was due to cytotoxic effects caused by 2A overexpression (Fig. 4f). These results suggest that EV71 2A protease causes the breakdown of the SETD3-

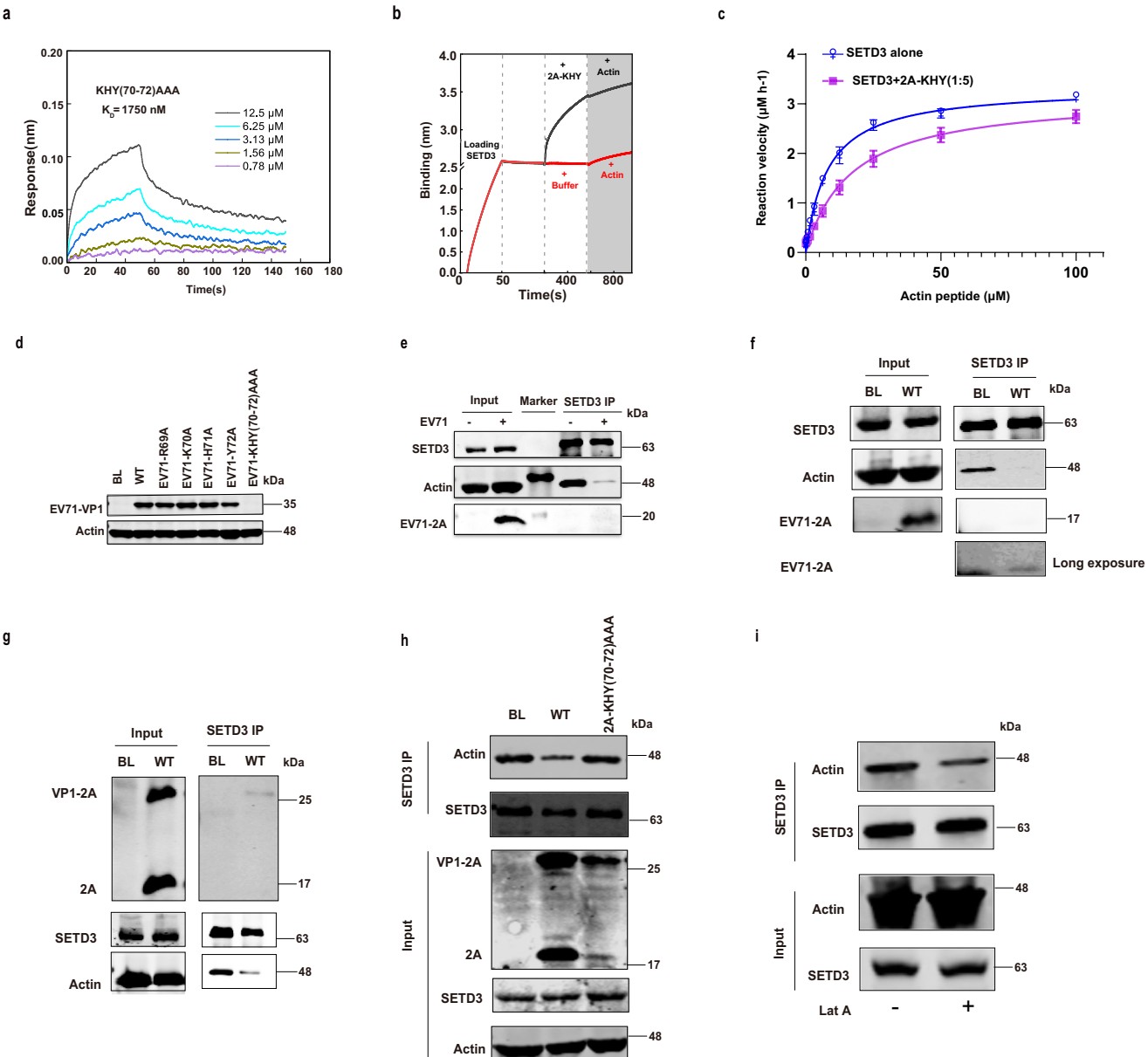

**Fig. 4 | Disruption of the SETD3-actin Complex in EV71-infected Cells and in EV71 2A-overexpressing Cells. a** BLI experiments of SETD3 1–498 for binding 2A KHY(70-72)AAA mutant. **b** BLI experiments illustrating preloading EV71 2A KHY(70-72)AAA mutant to SETD3 (immobilized on SA biosensor) was disrupted by subsequent loading of AP-actin. **c** The MTase activity of SETD3 is not inhibited by EV71 2A KHY(70-72)AAA mutant. Data are mean ± s.d. from three independent experiments (*n* = 3). Source data are provided as a Source Data file. **d** RD cells mock-infected (BL) or infected with supernatants from BSR/T7 cells transfected with the wild-type EV71 infectious clone (WT) and or mutants. Source data are provided as a Source Data file. Data are representative of three independent experiments. **e** RD cells mock-infected (-) or infected with EV71 (MOI = 10, +). Cell lysates (Input) and immunoprecipitates (SETD3 IP) were analyzed by western blotting to detect SETD3, actin, and EV71 2A using corresponding antibodies. Source data are provided as a Source Data file. Data are representative of three independent experiments. **f** BSR/T7 cells mock-transfected (BL) or transfected with plasmids encoding 2A. Cell lysates (Input) and immunoprecipitates (SETD3 IP) were analyzed by western blotting to detect the indicated proteins. Source data are provided as a Source Data file. Data are representative of three independent experiments. **g** BSR/T7 cells mock-transfected (BL) or transfected with plasmids encoding precursor VP1-2A. The proteins were detected using specific antibodies. Source data are provided as a Source Data file. Data are representative of three independent experiments. **h** At 48 h post-transfection, cells were examined using the method described in (**f**). The BSR/T7 cells were either mock-transfected (BL) or transfected with plasmids encoding precursor VP1-2A harboring the indicated mutations. Source data are provided as a Source Data file. Data are representative of three independent experiments. **i** BSR/T7 cells mock-transfected (BL) or transfected with plasmids encoding precursor VP1-2A. Cell were not treated (−) or treated (+) with 0.5 μM latrunculin A (Lat-A) for 3 h, then analyzed as described in (**f**). Source data are provided as a Source Data file. Data are representative of three independent experiments.

actin complex during virus infection. It is worth noting that although the destruction of SETD3-actin complex was evident, the interaction between SETD3 and EV71 2A protease was barely detectable in EV71-infected cells or 2A overexpressed cells. Only a faint band was visible in the SETD3 Co-IP experiment of the 2A overexpressing cells after long exposure to enhance the visibility (Fig. 4e, f). This result suggests that the SETD3-2A complex is probably transient in the infected cells. An alternative possibility is that only a minor portion of SETD3 binds to 2A within cells, the sensitivity of co-immunoprecipitation experiments may be insufficient to detect this interaction.

Next, we devised another overexpression system capable of detecting 2A protease activity and its ability to disrupt the SETD3-actin interaction simultaneously, which is useful to understand the correlation between the disruption of SETD3-actin complex and virus replication. In this system, a precursor containing a C-terminal portion of VP1 linking to intact 2A region (VP1-2A) was expressed as previously described[26]. Briefly, when overexpressed, VP1-2A is initially transcribed, followed by the production of 2A via autocleavage. Expressing VP1-2A allowed the detection of its autocleavage products, which could serve as an indicator of the protease activity of 2A. First, we examined whether this system could function as intended. The result shows that the overexpression of VP1-2A also disrupted SETD3-actin complex resembling the effects caused by overexpression of 2A. Importantly, the mature 2A resulted from VP1-2A autocleavage could be detected in the cell lysates (Fig. 4g). Next, we detected the presence of the SETD3-actin complex in cells expressing 2A mutants that abrogated SETD3 binding and were lethal to viral propagation: 2A-KHY (70-72) AAA (Fig. 4h). As anticipated, 2A-KHY (70-72) AAA failed to disrupt the SETD3-actin complex in cells. Nevertheless, 2A-KHY (70-72) AAA still had residual proteinase activities (Fig. 4h). This result underscores the importance of disrupting the SETD3-actin complex during viral infection.

Combining the above results led us to postulate that the 2A-mediated SETD3-actin complex disruption might cause F-actin depolymeration. To test this hypothesis, we used an actin polymerization inhibitor latrunculin A (Lat-A) to treat normal cells. As shown in Fig. 4i, our Co-IP experiments show that Lat-A treatment attenuated the signal of actin in the immunoprecipitates, similar to the effects caused by 2A overexpression, indicating 2A-mediated SETD3-actin complex disruption might cause F-actin depolymeration.

In summary, our results provide evidence for key step in EV71 replication involving 2A protease, 2A occupies the central cleft of SETD3, thereby disrupting the SETD3-actin complex through competitive binding. Disruption of the SETD3-actin complex is important for EV replication.

## Methods

### Cells and viruses

Rhabdomyosarcoma (RD) cells were purchased from ATCC and cultured in modified Eagle's medium (MEM) supplemented with 10% fetal bovine serum (FBS) and 100 mg/ml penicillin and streptomycin. BSR/T7 cells were described in our previous studies[13,27] and cultured in Dulbecco's modified Eagle's medium (DMEM) supplemented with 10% FBS and 1 mg/ml G418. All cells were maintained at 37 °C in a humidified atmosphere of 5% $CO_2$ and 95% air. The EV71 Fuyang strain (GenBank accession no. FJ43976 9.1) was propagated in RD cells(ATCC). None of the cell lines utilized in this study were found in the International Cell Line Authentication Committee database. All cell lines underwent monthly testing to confirm the absence of mycoplasma contamination. The cell lines were newly thawed from the original seed stocks and cultured for a maximum of one month. Prior to experimentation, the cell line morphology was verified and compared against ATCC reference images to prevent any potential cross-contamination or misidentification issues.

### Antibodies and reagents

The following antibodies were used in this study: β-actin (A1978; Sigma; Dilution 1:5000), V5 (V8012; Sigma; Dilution 1:1000), EV71-VP1 (MAB1255-M05; Abnova; Dilution:1:1000), and SETD3 (ab176582; Abcam; Dilution 1:2000). IRDye 680- and 800-labeled secondary antibodies were purchased from LI-COR Biosciences (926-68020, 926-32211, 926-68073; Dilution: 1:5000-1:20000). Anti-EV71 2A antibody was generated in rabbits using recombinant protein as the immunogen (Dilution 1:5000). Latrunculin A (L5163) was purchased from Sigma.

### Plasmid construction and the EV71 infectious clone

Genes encoding EV71 2A and SETD3 were chemically synthesized by Sangon Biotech (Shanghai, China) and codon-optimized for expression in insect cells and 293T cells (Supplementary Data 1). For cryo-EM, the *SETD3* and EV71 *2A* genes were amplified by PCR using primers listed in Supplementary Data 2 and cloned into a pFastBac Dual vector (Invitrogen) for coexpression. The gene for *SETD3* was inserted into open reading frame 1 (ORF1; between *Bam*HI and *Hind*III restriction sites) containing an N-terminal His-tag and a tobacco etch virus (TEV) protease cleavage site, and the gene for EV71 *2A* was inserted into ORF2 (between *Xho*I and *Kpn*I restriction sites) without a tag. The human β-*actin* gene was synthesized and optimized for expression in sf21 cells by Genscript (Supplementary Data 1), and then amplified and cloned into the pFastbac-HTB vector using the NcoI and XhoI restriction sites (Supplementary Data 2). This vector includes a Tobacco Etch Virus (TEV)-cleavable N-terminal His tag. To prevent actin polymerization, we introduced the A204EP243K mutation site as described previously[23]. Briefly, we initiated the construction of the A204E mutation following the instructions provided in the Q5 Site-Directed Mutagenesis Kit (NEB #E0554S). After conducting PCR with A204E mutant primers, the Kinase Ligase-DpnI (KLD) enzyme mix was added to circularize the PCR products and remove the template. Subsequently, the plasmid was transformed into chemically-competent *E. coli*. The sequences of the A204E mutant were confirmed using DNA sequencing. The same experimental procedure was carried out for the construction of the P243K mutant. For crystallography, genes encoding EV71 2A and SETD3(1−498) were separately subcloned into pET-28a and/or the pET-28a-SUMO vector to produce target proteins with an N-terminal His-tag (pET-28a-N-His-EV71-2A-C110A) and an N-terminal 6×His-SUMO tag (pET-28a-N-His-SUMO-SETD3[1−498]) as previously described[28]. Briefly, the gene encoding SETD3[1-498] was inserted into a pET-28a-SUMO vector between the BamHI and XhoI sites, expressing SETD3[1-498] with an N-terminal 6×His-SUMO tag. Plasmids expressing EV71 2A mutants were constructed using site-directed mutagenesis (QuickChange).

The pcDNA3.1-IRES-2A construct was a generous gift from Dr. Shih-Yen Lo (Tzu Chi University, Taiwan, China). Plasmids expressing 2A and VP1-2A mutants were generated by site-directed mutagenesis using pcDNA3.1-IRES-2A as the template. The EV71 infectious clone was a generous gift from Dr. Shan Cen (Institute of Medicinal Biotechnology, Chinese Academy of Medical Sciences, Beijing, China). Mutated infectious clones were generated by PCR amplification using the WT EV71 infectious clone as a template.

### Protein expression and purification

For cryo-EM, SETD3(residues 1−503) constructs and EV71 2A C110A mutants were coexpressed in Sf21 cells (Invitrogen) using the Bac-to-Bac Baculovirus Expression System (Invitrogen). A 2 l culture of sf21 cells was infected with 60 ml recombinant baculovirus at 28 °C. After 48 h of recombinant baculovirus infection, insect cells were collected by centrifugation, and the cell pellet was resuspended in lysis buffer (20 mM Tris-HCl pH 8.0, 100 mM NaCl, 10 mM imidazole, 10 mM β-mercaptoethanol, and 1 mM PMSF) and lysed by sonication for 20 min. The lysate was centrifuged, and the supernatant was collected; Ni-NTA resin was applied to a gravity column pre-equilibrated with lysis buffer, and the sample was loaded onto the column. The resin was washed three times with buffer containing 20 mM Tris-HCl pH 8.0, 100 mM NaCl, and 20 mM imidazole. Fractions containing the target protein were eluted with elution buffer containing 20 mM Tris-HCl pH 8.0, 100 mM NaCl, and 300 mM imidazole. Subsequently, the N-terminal His-tag was cleaved from the target protein by adding TEV protease during overnight dialysis against buffer containing 20 mM Tris-HCl pH 8.0 and 100 mM NaCl. The sample was reloaded onto Ni-NTA resin to remove the His-tag; the flow-through containing cleaved target protein was collected and diluted with buffer containing

20 mM Tris-HCl pH 8.0, and further purified by cation-exchange chromatography using a HiTrap Q HP column (GE Healthcare) equilibrated with buffer containing 20 mM Tris-HCl pH 8.0 and 75 mM NaCl, and eluted with a linear gradient from 10 mM to 1000 mM NaCl. Finally, the flow-through fraction containing the target protein was purified by gel-filtration chromatography on a Superdex 200 10/300 column (GE Healthcare) using buffer containing 20 mM Tris pH 8.0 and 150 mM NaCl.

For *E. coli* expression, each of the constructs was separately transformed into the *E. coli* BL21 (DE3) strain, and cells were cultured in Luria broth (LB) at 37 °C overnight. For large-scale expression, when bacterial cultures reached an optical density at 600 nm ($OD_{600}$) of 0.6, they were induced by adding isopropyl-β-D-thiogalactopyranoside (IPTG) to a final concentration of 0.5 mM, and culturing was continued at 16 °C for 16–20 h. The purification procedures described above were used to purify SETD3(1–498) and EV71 2A C110A, except that the EV71 2A His-tag was removed by thrombin protease, and the SETD3(1–498) SUMO tag was removed by Ulp1 peptidase. To assemble the SETD3(1–498) and EV71 2A C110A complex, separately purified 2A C110A and SETD3(1–498) were combined at a molar ratio of 1:3, incubated at 4 °C overnight, and purified by size-exclusion chromatography using a Superdex 200 10/300 column (GE Healthcare) pre-equilibrated with storage buffer containing 20 mM Tris pH 8.0 and 150 mM NaCl.

For human β-actin expression and purification. The final plasmid was transformed into DH10Bac competent cells, and the recombinant proteins were expressed in insect cells using the Bac-to-Bac Baculovirus Expression System (Invitrogen). The cells were harvested by centrifugation at 3,470 ×g for 15 min and lysed in a solution containing 1 M Tris-HCl pH 8.0, 100 mM KCl, 0.5 mM $MgCl_2$, 0.5 mM ATP, 4% Triton X-100, and 1 mM PMSF. Cell debris was removed by centrifugation at 47,850 × *g*, 4 °C for 1 h. The supernatant was loaded onto Ni-NTA resin (Qiagen) by gravity. Non-specific binding proteins were removed using a wash buffer (10 mM Tris-HCl pH 8.0, 100 mM KCl, 0.5 mM $MgCl_2$, 0.5 mM ATP, 20 mM imidazole). The target protein was stripped by elution buffer (10 mM Tris-HCl pH 8.0, 100 mM KCl, 0.5 mM $MgCl_2$, 0.5 mM ATP, 250 mM imidazole). Subsequently, the N-terminal His-tag was removed by TEV protease digestion in dialysis buffer (10 mM Tris-HCl pH 8.0, 100 mM KCl, 0.5 mM $MgCl_2$, 0.5 mM ATP) overnight at 4 °C. The mixture was re-loaded onto Ni-NTA resin to remove His tag. The flowthrough containing the non-tagged target protein was collected and concentrated to 5 ml, and then depolymerized using dialysis in G buffer (2 mM Tris-HCl pH 8.0, 0.2 mM $CaCl_2$, 0.2 mM ATP) for 3 days. Finally, the G-actin was concentrated and loaded onto the Superdex 200 10/300 GL column (GE Healthcare) pre-equilibrated with G buffer. Pooled fractions containing the G-actin were concentrated and stored at −80 °C for further use.

## Crystallization and structure determination

Crystallization of SETD3(1–498) in complex with EV71 2A C110A was carried out by mixing 1 μl of 10 mg/ml protein and 1 μl reservoir buffer containing 0.02 M citric acid, 0.08 M BIS-TRIS propane pH 8.8, and 16% PEG3350, using the hanging-drop vapor diffusion method at 20 °C. Single crystals appeared within 7 days and grew to full size after ~10 days. For cryoprotection, crystals were transferred to reservoir buffer supplemented with 20% ethylene glycol and flash-cooled in liquid nitrogen. Complete X-ray diffraction data were collected on beamline BL10U2 and BL19U1 at the Shanghai Synchrotron Radiation Facility (SSRF) at 100 K. Diffraction data were processed using the XDS package[29]. Molecular replacement was performed with Phaser MR in CCP4i GUI (PDB 3W95 and 6MBK served as search models)[30]. Models were then subjected to successive rounds of manual building and refinement in Coot and PHENIX[31]. Data collection and structure refinement statistics are summarized in Supplementary Table 1. Figures were prepared using Pymol.

## Cryo-EM sample preparation and data collection

A 3 μl sample of purified SETD3(1−503)-EV71 2A protein complex at 3.6 μM was applied to Quantifoil R1.2/1.3 Cu 300 mesh holey carbon grids that were glow-discharged at 50 W for 1 min in $H_2$-$O_2$ by Gatan Solarus (Gatan, USA). The grids were vitrified in liquid ethane using a Vitrobot Mark IV instrument (Thermo Fisher Scientific) at 100% humidity and 4 °C for a 7 s blot duration with a blot force of 4. Micrographs of the SETD3(1−503)-EV71 2A complex were collected at the Center for Biological Imaging, Core Facilities for Protein Science, Institute of Biophysics, Chinese Academy of Sciences. Specifically, high-resolution data were collected on a Titan Krios electron microscope (Thermo Fisher Scientific) running at a 300 kV accelerating voltage equipped with a K3 direct electron detector (Gatan) in bin 0.5, with a physical pixel size of 0.82 Å for each pixel. Each micrograph stack was dose-fractionated to 32 frames, yielding a total electron dose of 66 e$^-$/Å$^2$ and a total exposure time of 6 s. Data were obtained using SerialEM software[32] using a beam-image shift data collection strategy and a defocus value ranging from −1.8 to −2.5 μm[33].

## Cryo-EM data processing

Cryo-EM data were calculated by the Laboratory for Condensed Matter Physics, Institute of Physics, Chinese Academy of Sciences. All processes were performed using cryoSPARC-3.3.2[34]. All movie frames of the SETD3(1−503)-EV71 2A complex were translationally aligned and dose-weighted using Patch Motion Correction[35]. Patch contrast transfer function (CTF) estimation was used to correct the CTF parameters of dose-weighted micrographs[36]. A total of 2,211,651 particles were automatically picked and subjected to reference-free 2D classification. After several rounds of 2D and 3D classification, two distinct conformations of the SETD3(1−503)-EV71 2A complex among the 950,903 particles were identified, namely, class 1 (603,524 particles) and class 2 (347,379 particles), without EV71 2A binding. Class 1 exhibited high-resolution structural features, and particles were subjected to 3D nonuniform refinement, local refinement, and autosharpen, yielding final maps for conformation 1 with an overall resolution of 3.14 Å based on the gold-standard Fourier shell correlation (FSC) 0.143 criterion. Local resolution estimation with half-reconstructions was used to determine the local resolution of each map.

## Model building

The EV71 2A (PDB: 3W95) and SETD3 (6MBK) models were rigid body fitted into the cryo-EM map using UCSF Chimera and then refined in the PHENIX package[37,38]. The resulting model was then manually adjusted in Coot[39] iteratively, including improvement of main chains and side chains of residues, and addition of residues where the corresponding density showed density features. Validation of the model was performed using comprehensive validation (cryo-EM) in PHENIX. Figures were prepared in UCSF Chimera[37]. Cryo-EM data collection and refinement statistics are summarized in Supplementary Table 2.

## Size-exclusion chromatography

A Superdex 200 10/300 GL column (GE healthcare) precalibrated using gel filtration standards γ-globulin (158 kDa), ovalbumin (45 kDa), myoglobin (17 kDa), and vitamin B12 (1.35 kDa) in buffer containing 20 mM Tris pH 8.0 and 150 mM NaCl was used to analyze the molecular weight of purified SETD3 FL and/or SETD3(1−498) complexed with EV71 2A C110A proteins. The size-exclusion chromatography profiles of complexes were analyzed by GraphPad Prism software. Elution fractions from size-exclusion chromatography were further analyzed by SDS-PAGE and Coomassie Brilliant Blue staining.

## Analytical ultracentrifugation (AUC)

Analytical ultracentrifugation was performed using Beckman Optima XL-I equipped with an AN-50 Ti rotor. The 2A, SETD3 and 2A-SETD3 complex samples were freshly purified in a buffer (20 mM Tris pH = 8.0, 150 mM NaCl), then the protein samples were concentrated to absorbance 280 nm=0.8-1 before measurement. The differential sedimentation coefficients(S), c(s), frictional coefficients, and molecular weight were calculated by the XL-I data analysis software.

## Biolayer interferometry (BLI)

The binding affinity of SETD3 to 2A was assessed using Biolayer Interferometry (BLI) with the ForteBio Octet RED96e Analysis System. Briefly, purified SETD3 was initially biotinylated using a biotinylation kit (Frdbbio, ARL0020K). Subsequently, 10 μg/ml biotinylated SETD3 was immobilized onto SA biosensors for 200 s. The WT and different 2A mutants were then subjected to continuous dilutions in a buffer (50 mM HEPES pH 8.0, 100 mM NaCl) and incubated with the biosensors for 50 s. The biosensors were then transferred to disassociation for 100 s in fresh buffer to obtain a kinetics. In the competitive BLI assay, the Biotinylated SETD3 protein (10 μg/ml) was immobilized onto SA biosensors and immersed in wells with/without 20 μM actin in G buffer. The differences in binding curves were recorded with and without 20 μM 2A. The same experiment was performed by exchanging the sample position between actin and EV71 2A. Specifically, biotinylated SETD3 was initially incubated with/without 20 μM EV71 2A or KHY(70-72)AAA mutant in G buffer and then transferred to wells with or without 20 μM actin. Finally, all the biosensors were washed in fresh buffer. The binding affinity of EV71 2A to SETD3 with or without SAH was conducted by BLI. The Biotinylated EV71 2A (10 μg/ml) was immobilized onto SA biosensors, followed by immersion into wells containing 0.5 μM SETD3 pre-incubated with various concentrations of SAH. After association, all the sensors were transferred into a buffer (50 mM HEPES pH 8.0, 100 mM NaCl, 1% DMSO) to record dissociation kinetics. All BLI experiments were conducted at 25 °C. Data analysis was performed using ForteBio Data analysis v 11.1 software. The data was processed with reference well subtraction and Global fitting with a 1:1 model. Two independent experiments were performed for each sample.

## Methyltransferase assay

The steady-state kinetic assay was conducted using a short actin peptide (66-TLKYPIEHGIVTNWD-80) as the substrate for SETD3 as described previously[19]. Briefly, each reaction mixture contained 20 mM Tris-HCl (pH 8.0), 50 mM NaCl, 1 mM EDTA, 3 mM $MgCl_2$, 0.1 mg/ml BSA, 1 mM DTT, 0.18 μM SETD3, and 20 μM SAM with a total volume of 20 μl. The reactions were assessed using the MTase-Glo™ Methyltransferase Assay kit (Promega, Cat# V7601). Reactions were carried out for 20 min at room temperature, and then 0.5% trifluoroacetic acid was added to quench the enzyme reaction. The reaction mixtures were loaded into Greiner 96-well plates, and luminescence was measured using the SpectraMax® iD5 Multi-Mode Microplate Reader. Data analysis was performed using GraphPad Prism software using Michaelis–Menten equation.

MTase activity inhibition experiment by EV71 2A or KHY(70-72) AAA mutant followed the similar protocols as described above, with the addition of EV71 2A or the KHY(70-72)AAA mutant at 2-folds and 5-folds excess over SETD3 in the reaction mixtures prior to MTase activity measurement. All experiments were repeated three times and results are presented as mean values with s.d.

## Virus rescue from the EV71 infectious clone

The EV71 infectious clone or infectious clones bearing 2A mutations were transfected into BSR/T7 cells. At 48 h post-transfection, supernatants were collected by centrifugation at 2000 g for 10 min at 4 °C.

Supernatants were used to infect RD cells previously seeded in 12-well plates. At 12 h postinfection, RD cells were harvested and lysed. EV71 virus production was analyzed by western blotting to detect EV71 VP1 expression.

## Coimmunoprecipitation

Cells transfected with pcDNA3.1-IRES-2A /2A mutants expressing plasmids or infected with EV71 virus were lysed on ice for 30 min in lysis buffer (25 mM Tris, 150 mM NaCl, 1 mM EDTA, 1% NP-40, pH 7.4). Equal amounts of total proteins were incubated overnight with SETD3 antibody (1:300) at 4 °C. The following day, protein A agarose beads (P2545; Sigma) were washed and added to cell lysates and incubated for an additional 4 h at 4 °C. Immunoprecipitates were washed three times with lysis buffer. Finally, agarose beads were boiled in 2× SDS sample buffer and analyzed by western blotting.

## Western blotting

Equal amounts of cell lysates or immunoprecipitates were subjected to 10% or 12% SDS-PAGE then transferred to a nitrocellulose membrane (Pall). After blocking with 5% nonfat milk in TBS at room temperature for 1 h, membranes were incubated with corresponding primary antibodies overnight at 4 °C, followed by incubation with IRDye-labeled secondary antibodies at room temperature for 2 h. Finally, membranes were scanned with an Odyssey infrared imaging system (Li-COR Biotechnology). The uncropped and unprocessed scans of all the gels and or blots are provided in the Source Data file.

## Statistics and reproducibility

All results are shown in Figs. 3e and 4c were repeated at least three times, and data are mean ±s.d. from three independent experiments (n = 3); All results shown in Fig. 1b-h, Fig. 3f, g, Fig. 4a, b, d-I and supplementary Figs. 2, 7, and 8 were repeated at least three times and representative images are shown in this paper. The experiments were not randomized. The Investigators were not blinded to allocation during experiments and outcome assessment.

## Reporting summary

Further information on research design is available in the Nature Portfolio Reporting Summary linked to this article.

# Data availability

The 3D cryo-EM density map of the SETD3(1-503)-EV71 2A complex has been deposited in the EM Database under the accession codes EMD-38156, and the coordinates for the structure have been deposited in Protein Data Bank under accession code 8X8Q. The atomic coordinates and structure factors for SETD3(1-498)-EV71 2A complex have been deposited in the Protein Data Bank under the accession codes: 8X77. Publicly available protein atomic models with the following PDB code were used in the study: 6MBK, 3W95, 5OOF, 7LMS, 6OX2, 3SMT and 4FVD. Source data are provided with this paper.

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

## Acknowledgements

We thank X. Huang, T. Niu, B. Zhu, X. Li, L. Chen and other staff members at the Center for Biological Imaging (CBI), Core Facilities for protein Science at the Institute of Biophysics, Chinese Academy of Science (IBP, CAS) for the support in cryo-EM data collection. We thank the staff of BL19U1 beamline of the National Facility for Protein Science in Shanghai (NFPS), the staff of the BL17U1 beamline and the staff of the BL10U2 beamline at the Shanghai Synchrotron Radiation Facility for assistance in data collection. We thank Dr. Qing Chang for AUC data collection at the Protein Preparation and Characterization Platform of the Tsinghua University Technology Center for Protein Research. We thank the staff from the Core Facility of National Institute of Pathogen Biology, Chinese Academy of Medical Sciences. This work was supported by National Key Research and Development Program of China (2023YFC2307803); Chinese Academy of Medical Sciences (CAMS) Innovation Fund for Medical Sciences (2022-I2M-1-021); the Non-profit Central Research Institute Fund of Chinese Academy of Medical Sciences (2023-PT310-04); National Natural Science Foundation of China (82341095; 82261160398; 81971985; 82272308;82221004;81930063); the Fundamental Research Funds for the Central Universities (3332021092; 3332023165).

## Author contributions

S.C., J.W., and X.G. designed the study. S.C., W.D. and X.G. solved the X-ray and EM structures. S.C., X.G. and B.W. wrote the paper. X.G., B.W., K.Z., L.W., K.S and B.Q. performed experiments, and analyzed the data. All authors reviewed the results and approved the final version of the manuscript.

## Competing interests

The authors declare no competing interests.
