## [Peer Review File · Nature Communications]

The EV71 2A protease occupies the central cleft of SETD3 and disrupts SETD3-actin interactionReviewer #1 (Remarks to the Author):

This manuscript describes biostructural analyses of SETD3 in complex with enterovirus 71 2A protease using x-ray crystallography and cryo-electron microscopy. SETD3 belongs to a family of biomedically important methyltransferases that catalyze methylation of lysine, arginine and histidine residues in various proteins. The authors have investigated how does the actin-specific methyltransferase SETD3 interact with EV71 2A protease, the association important for virus replication. Along with structural analyses, the authors carried out mutagenesis, binding analyses and immunoprecipitation studies that collectively revealed that EV21 2A outcompetes actin for binding to SETD3. The manuscript is well written and of good quality. The work will be appealing to chemical biologists and biochemists, in particular those investigating molecular mechanisms of protein-protein interactions.

Thermodynamic analyses showed that SETD3 binds stronger CVB3 2A than related EV71 2A and HRV-C15 2A proteases. The authors need to describe plausible reasons for such differences, in particular because key residues in EV71 2A for strong association with SETD3 have been identified and related SETD3-CVB3 2A structure has been reported (ref. 21, PDB: 7LMS). Some additional analyses on comparisons between both structures are recommended, both in the main text and Supporting Information.

The quality of ITC data is generally poor (see Figure 3). The heats released or gained are very low, which makes fitting curves rather poor and thermodynamic parameters possibly incorrect. ITC experiments could be repeated at higher concentrations of SETD3 and 2A to obtain better thermodynamic data that are important for the narrative of this manuscript. Even better, the authors could carry out biolayer interferometry assays, which provide an orthogonal binding method for such biomolecular systems (see ref. 21).

In competition assays, two sequences of actin were used. Only the shorter, less relevant, sequence was shown to be outcompeted by EV71 2A. Binding affinities for SETD3-actin(66-88) and SETD3-EV71 2A are basically indistinguishable (1.23 μ M vs 1.06 μ M). These results indicate that full length EV71 2A has similar binding affinity as a 23-mer actin peptide fragment, and one could envision that full length actin protein would bind even stronger. If this is the case, the conclusion that EV71 2A protease outcompetes actin for binding with SETD3 would be wrong. The authors could carry out additional binding work with full length actin to conclusively demonstrate the superiority of EV71 2A over actin for binding with SETD3.

For readers, the natural substrates and the active site of EV71 2A protease could be briefly described. Is the active site accessible in the SETD3-EV71 2A complex? Although the inactive variants were used in this study, the structure would still provide an insight into the location of the 2A protease active site.

Figure 1C shows the SEC elution profile and SDS-PAGE data of the SETD3-EV71 2A complex. The authors state that binding is in 1:1 ratio. The experimental proof for this statement needs to be provided.

More detailed structural analyses are recommended in Figure 3. Some ITC data can be moved from Figure 3 to Supporting Information.

Reviewer #2 (Remarks to the Author):

In this manuscript, the authors analyzed the structures of SETD3 in complex with viral protease 2A, by x-ray crystallography and cryo-EM. The structures are similar to those recently published by another group in Nature Communications (PMID 36075902), with additional concerns (see below). The functional study does not go beyond what is already known that the host factor SETD3 is important in enteroviral replication, but instead it adds confusion.

The structure-and-function study reached conflicting conclusions: e.g. stable complexes of SETD3-

2A can be formed and purified from *E. coli* or co-expressed in insect cells, yet the authors claims SETD3-2A complex is probably transient or partially occupied or barely detectable or a minor portion of SETD3 binds to 2A in infected cells; yet 2A is sufficient to disrupt SETD3-actin complexes, and yet 2A did not coprecipitate with SETD3, all conclusions difficult to reconcile. It is obvious that the authors made many speculative statements.

1. The most interesting part of the data is that the binding affinity (KD) of SETD3 with CVB3 2A is ~50x and ~130x stronger than that of EV71 2A and HRV-C15 A2, respectively (21 nM vs. 1 uM vs. 2.65 uM). Does this imply that viral production of coxsackievirus B3 (CVB3) in cells is much more than the other two viruses? (as the authors indicated in the Abstract that the binding ability between SETD3-A2 correlates with viral production in cells).

There are no error bars or statistical significance provided for the ITC KD measurements?

2. No reason was given why CVB3 2A (the strongest binder) was not chosen for further study.

3. Since the A2 sequences are highly similar and residues mediating interactions are invariant (Figure 3C), what are the factors contributing to the 50x-130x stronger binding affinity by CVB3 2A?

4. The competition assays between actin and A2 bindings with SETD3 are not convincing. First, the two binding areas between EV71 2A and SETD3 are very small, 728A2 and 240A2. Although not specifically provided in the current manuscript, I would suggest the authors to calculate the interface between SETD3 and actin peptide (residues 66-88) from the established structures. The competition assays between 2A protein and actin peptide are not a direct comparison (i.e., a folded protein vs. a peptide).

5. If the SETD3-2A interaction is dynamic or transient, does 2A affect methylation activity of SETD3 on actin substrate? If SETD3 is still active on actin in the presence of A2, it will be more meaningful to measure the change of Km value from the kinetics study.

The premise of the study seems to suggest that SETD3 and actin form a stable and long-lasting complex. However, what could be the reason(s) for the enzyme to form a stable complex with substrate (or product)?

6. 2A Y94 was identified as the key residue for interaction with SETD3; does the SETD3 F409 mutant (the residue makes interaction with 2A Y94) has the same effect?

7. Does the cofactor SAM or SAH has any effect SETD3-2A interaction?

8. The difference seen between X-ray and cryo-EM models are probably due to the nature of low resolution.

9. How did the authors reach the conclusion of 1:1 ratio of SETD3 and A2 in Figure 1C?

10. What is the zinc finger of 2A? why is it involved in protein-protein interactions? Any references on the statement?

11. The conclusion – SETD3-2A involves more van der Waals and hydrophobic contacts – is not supported by the description. EV71 2A H71-SETD3 Y288, EV71 2A R115-SETD3 S264, and EV71 2A Q117-SETD3 Q257, EV71 2A K70-SETD3 E296, residues R115 and Q117, SETD3 Q380 and EV71 2A K78 are all charged and polar residues.

Q380 salt bridges with K78 is incorrect. The salt bridge only applies to charge-charge interactions, and Q380 is not a charged residue.

Among all the residues being discussed, only four residues F409, L81, Y83 and Y94 are hydrophobic and aromatic residues.

The functional significance of Y94 cannot be appreciated from the current structure description.

11. The remarkable conformational rearrangement of R-strand is not clear from Fig. 3D.

12. Is H71 of 2A located in the same binding site of actin histidine substrate? The details of H71 binding through backbone hydrogen bonds should be shown.

Point-to-point responses to reviewers' comments

REVIEWER COMMENTS

Reviewer #1 (Remarks to the Author):

This manuscript describes biostructural analyses of SETD3 in complex with enterovirus 71 2A protease using x-ray crystallography and cryo-electron microscopy. SETD3 belongs to a family of biomedically important methyltransferases that catalyze methylation of lysine, arginine and histidine residues in various proteins. The authors have investigated how does the actin-specific methyltransferase SETD3 interact with EV71 2A protease, the association important for virus replication. Along with structural analyses, the authors carried out mutagenesis, binding analyses and immunoprecipitation studies that collectively revealed that EV21 2A outcompetes actin for binding to SETD3. The manuscript is well written and of good quality. The work will be appealing to chemical biologists and biochemists, in particular those investigating molecular mechanisms of protein protein interactions.

Responses:

We appreciate these comments.

Thermodynamic analyses showed that SETD3 binds stronger CVB3 2A than related EV71 2A and HRV-C15 2A proteases. The authors need to describe plausible reasons for such differences, in particular because key residues in EV71 2A for strong association with SETD3 have been identified and related SETD3-CVB3 2A structure has been reported (ref. 21, PDB: 7LMS).

Some additional analyses on comparisons between both structures are recommended, both in the main text and Supporting Information.

Responses:

We believe more experiments are needed to address this question.

(1) Because Peters et al (Nat Commun 13: 5282, 2022) used biolayer interferometry (BLI) to measure the SETD3-CVB3 2A binding, we also established BLI assays to investigate binding affinity of SETD3 for EV71 2A and compare it with SETD3-CVB3 2A binding, news results are illustrated in the Figure below.

BLI results indicate that the EV71 2A and CVB3 2A exhibited rather similar binding affinity, with the K_d 22.7 nM and 11.8 nM, respectively. New results have been added to revised Fig.1 f-g to replace old ITC results. We give up old ITC results because they are of low quality and provide confusing results. Description of these results are included in main text, Page 6, Line 113-120.

Our previous ITC results along with conclusions based on ITC were deleted in the revised paper.

(2) We superimposed the EV71 2A-SETD3 structure with the CVB3 2A-SETD3 structure (PDB: 7LMS), which gave an overall rmsd values of 1.44 Å with 607 Ca atoms aligned (Figure below). Our structural alignment shows that binding mode of SETD3 with different 2A is essentially similar, see Figure below. We added these new results to the revised manuscript. (page7-8, line154-159)

(3) We further compared residues at the SETD3-2A interface (Figure below), demonstrating that residues of two viral 2A protease involved in the interaction with SETD3 are highly conserved.

We added new structural comparison analyses in the revised Fig.2d and Fig.S5. We added these new results to the revised manuscript. (page7-8, line154-159)

The quality of ITC data is generally poor (see Figure 3). The heats released or gained are very low, which makes fitting curves rather poor and thermodynamic parameters possibly incorrect. ITC experiments could be repeated at higher concentrations of SETD3 and 2A to obtain better thermodynamic data that are important for the narrative of this manuscript. Even better, the authors could carry out biolayer interferometry assays, which provide an orthogonal binding method for such biomolecular systems (see ref. 21).

Responses:

We agreed with this reviewer that ITC experiments should be repeated at higher protein concentrations in order to obtain better thermodynamic data. However, recombinant EV71 2A tended to precipitate in the syringe of the ITC instrument at concentrations above 0.5 mM, thus jeopardizing ITC titrations.

We therefore used BLI assays to investigate SETD3-2A binding. Given Peters et al (Nat Commun 13: 5282, 2022) also used BLI to measure SETD3- CVB3 2A binding, it would be appropriate to use the same technique to compare binding of SETD3 with different 2A proteins. We then established BLI experiments and re-measured binding SETD3-2A binding kinetics. We replaced the ITC results with new BLI results throughout the revised paper.

In competition assays, two sequences of actin were used. Only the shorter, less relevant, sequence was shown to be outcompeted by EV71 2A. Binding affinities for SETD3-actin(66-88) and SETD3-EV71 2A are basically indistinguishable (1.23 μ M vs 1.06 μ M). These results indicate that full length EV71 2A has similar binding affinity

as a 23-mer actin peptide fragment, and one could envision that full length actin protein would bind even stronger. If this is the case, the conclusion that EV71 2A protease outcompetes actin for binding with SETD3 would be wrong. The authors could carry out additional binding work with full length actin to conclusively demonstrate the superiority of EV71 2A over actin for binding with SETD3.

Responses:

As suggested, we prepared FL-human β -actin for our new competition assays. We first produced monomeric actin expressed in sf21 cells following a protocol described previously¹(see SDS-PAGE below).

To prevent actin self-polymerization, two mutations (A204E/P243K) were introduced to yield monomeric actins or AP-actin. As shown in the SDS-PAGE above, the His-tagged AP-actin was cleaved by TEV protease and the untagged AP-actin was collected in flowthrough, and the final product was further depolymerized via dialysis in G buffer (2 mM Tris-HCl pH 8.0, 0.2 mM CaCl_2 , 0.2 mM ATP) for 3 days to ensure constant presence of the nonpolymerizable actin monomers.

Next, we used AP-actin in the BLI-based competition experiments. We demonstrate (Figure below, left) that preloading EV71 2A to SETD3 cannot be disrupted by subsequent loading of actin. Black curve: SETD3 loaded on SA biosensor was first exposed to EV71 2A and subsequently AP-actin; Red curve: SETD3 loaded biosensor was first exposed to G buffer and subsequently AP-actin.

However, (Figure below, right) preloading AP actin with SETD3 had little effect on subsequent binding of EV71 2A. Black curve: SETD3 loaded on biosensor was first exposed to AP-actin and subsequently EV71 2A; Red curve: SETD3 loaded biosensor was first exposed to G buffer and subsequently EV71 2A. These results provide evidence that 2A can disrupt the SETD3-actin interaction via competitive binding. We included these results in the revised figure 3 f-g and in main text. (page 11, line235-245)

For readers, the natural substrates and the active site of EV71 2A protease could be briefly described. Is the active site accessible in the SETD3-EV71 2A complex? Although the inactive variants were used in this study, the structure would still provide an insight into the location of the 2A protease active site.

Responses:

Thanks for the advice. We added a magnified view of EV71 2A protease active site in the revised figure 3, A protease substrate peptide (magenta) is modeled in the active site by superimposing an EV71 2A-peptide complex (PDB: 4FVD) with the EV71 2A chain in our complex; catalytic triads are shown with stick models and colored blue. From the modeled structure, it is clear that the 2A active site is accessible to substrate in the SETD3-EV71 2A complex (Figure below). We also describe these analyses in main text (page 9, line195-199).

Figure 1C shows the SEC elution profile and SDS-PAGE data of the SETD3-EV71 2A complex. The authors state that binding is in 1:1 ratio. The experimental proof for this statement needs to be provided. More detailed structural analyses are

recommended in Figure 3. Some ITC data can be moved from Figure 3 to Supporting Information.

Responses:

To determine the stoichiometry of SETD3-EV71 2A complex, we carried out analytical ultracentrifugation (AUC) to calculate molecular mass. From the results (see figure below, panel e), we are confident that the SETD3-EV71 2A complex has 1:1 molar ratio. In detail, the calculated molecular mass of EV71 2A alone is ~17 kDa by AUC; the calculated molecular mass of SETD3 (1-498) is ~54 kDa by AUC, matching their theoretical molecular mass (16 kDa for 2A and 57 kDa for SETD3). The calculated molecular mass of the SETD3-EV71 2A complex is ~66 kDa by AUC, it is very close to its theoretical molecular mass, 73 kDa. We added these new results to the revised manuscript (page 5, line 98-104).

We added more detailed structural analyses in revised Figure 2d,3c,3d in the revised manuscript.

Reviewer #2 (Remarks to the Author):

In this manuscript, the authors analyzed the structures of SETD3 in complex with viral protease 2A, by x-ray crystallography and cryo-EM. The structures are similar to those recently published by another group in Nature Communications (PMID 36075902), with additional concerns (see below). The functional study does not go beyond what is already known that the host factor SETD3 is important in enteroviral replication, but instead it adds confusion.

Responses:

We thank this reviewer for the comments. Presently, there is no evidence to explain the precise molecular mechanism of how the interplay between the EV71 2A protease and SETD3 operates during the viral replication. Although the protease activity of 2A is important for EV replication², the activity has been demonstrated to be unrelated to SETD3-2A interaction^{3,4}. Moreover, the viral infection does not necessitate the methyltransferase activity of SETD3⁴. In revised manuscripts, we further demonstrated that the prebound of 2A to SETD3 blocked the binding of β -actin protein. These results

provide evidence that 2A can disrupt the SETD3-actin interaction via competitive binding in vitro; thus, 2A and actin cannot simultaneously bind with SETD3. More importantly, we further demonstrated that 2A efficiently disrupts the SETD3-actin complex either EV71-infected or 2A expressing cells using Coimmunoprecipitation, and the disruption of SETD3-actin complex undermines EV replication. our study uncovers the molecular mechanism underlying the interplay among SETD3, actin, and viral 2A during virus replication, and 2A-mediated disruption of SETD3-actin interaction is a key step during virus replication.

The structure-and-function study reached conflicting conclusions: e.g. stable complexes of SETD3-2A can be formed and purified from E. coli or co-expressed in insect cells, yet the authors claims SETD3-2A complex is probably transient or partially occupied or barely detectable or a minor portion of SETD3 binds to 2A in infected cells; yet 2A is sufficient to disrupt SETD3- actin complexes, and yet 2A did not coprecipitate with SETD3, all conclusions difficult to reconcile. It is obvious that the authors made many speculative statements.

Responses:

We apologize for the confusions, and we have thoroughly revised our manuscript. More experiments have carried out according to reviewers' comments. We revised many speculative statement. I hope the revised paper is clearer.

1. The most interesting part of the data is that the binding affinity (KD) of SETD3 with CVB3 2A is ~50x and ~130x stronger than that of EV71 2A and HRV-C15 A2, respectively (21 nM vs. 1 uM vs. 2.65 uM). Does this imply that viral production of coxsackievirus B3 (CVB3) in cells is much more than the other two viruses? (as the authors indicated in the Abstract that the binding ability between SETD3-A2 correlates with viral production in cells).

Responses:

Reviewer #1 has similar questions. We reasoned those large discrepancies in binding affinity between SETD3 and viral 2A was probably due to poor binding conditions. We were questioned by reviewer #1, that our previous ITC data has rather poor quality and the thermodynamic parameters are possibly incorrect (see our responses above). This reminded us that during our ITC titration, we did observe some protein precipitations in ITC syringe.

In order to make direct comparison with that reported in the Nat Commun (PMID 36075902) paper, we also established BLI experiments as suggested. We measured binding affinity of SETD3 for EV71 2A and CVB3 2A (Figure below). New results indicate that the EV71 2A and CVB3 2A exhibited rather similar binding affinity (22.7 nM and 11.8 nM). CVB3 2A has merely 2-folds higher affinity than EV71 2A for binding SETD3. Our previous ITC results along with conclusions based on ITC were deleted in the revised paper.

These results are supported by our structural analysis demonstrating: the mode of SETD3-2A interaction is highly similar and SETD3 binds a conserved region on 2A (Figure below). We included new BLI results in the revised Figure 1f-g and further structural analyses in the revised Figure 2d.

We added these new results to the revised manuscript (page 6, line 113-120; pages 7-8, line 154-159).

There are no error bars or statistical significance provided for the ITC KD measurements?

Responses:

We give up all ITC results and replace them with BLI results in the revised manuscript.

2. No reason was given why CVB3 2A (the strongest binder) was not chosen for further study.

Responses:

In our revised paper, we performed BLI experiments and found that EV71 2A and CVB3 2A actually has similar binding affinity with SETD3. Considering availability of virus strains, antibodies and infection clones, we chose EV71 as the model organism and focused on SETD3-EV71 2A interactions in this study. We add one sentence to explain our choice of model organism, page 6 line 128-130.

3. Since the A2 sequences are highly similar and residues mediating interactions are invariant (Figure 3C), what are the factors contributing to the 50x-130x stronger binding affinity by CVB3 2A?

Responses:

In the revised manuscript, we performed BLI assays as reviewer #1 suggested to re-assess the binding affinity of SETD3 for EV 2A and CVB 2A proteinases. The results indicated that the EV 2A and CVB 2A proteinases exhibited similar binding affinity (22.7nM for EV2A and 11.8nM for CVB 2A). we re-write this section in the revised manuscript. (Page6, line 113-120)

4. The competition assays between actin and A2 bindings with SETD3 are not convincing. First, the two binding areas between EV71 2A and SETD3 are very small, 728Å² and 240Å². Although not specifically provided in the current manuscript, I would suggest the authors to calculate the interface between SETD3 and actin peptide (residues 66-88) from the established structures. The competition assays between 2A protein and actin peptide are not a direct comparison (i.e., a folded protein vs. a peptide).

Responses:

We calculated interface area between SETD3 and actin peptide, ~1069 Å², which is slightly larger than that between SETD3 and EV71 2A (~969 Å², Site-1 plus Site-2 combined). However, it is insufficient to judge which interaction is stronger depending on interfacial area comparison. Especially, the interface area calculated from the actin peptide-bound SETD3 structure cannot fully represent interaction between SETD3 and a folded actin protein. We added description to the revised manuscript. (page10, line203-208)

We therefore prepared full length human β-actin for competition assays. We successfully produced monomeric actin expressed in sf21 cells following a protocol described previously ¹(Figure below).

To prevent actin self-polymerization, two mutations (A204E/P243K) were introduced to yield monomeric actin, denoted AP-actin. As shown in the SDS-PAGE below, the His-tagged AP-actin was cleaved by TEV protease and the untagged AP-actin was collected in flowthrough, and the final product was further depolymerized using dialysis in G buffer (2 mM Tris-HCl pH 8.0, 0.2 mM CaCl₂, 0.2 mM ATP) for 3 days to ensure constant presence of the nonpolymerizable actin monomer.

Next, we used monomeric AP-actin in the BLI assays of 2A and actin competing for binding with SETD3. We replaced here our old ITC results with BLI results for consistency with the BLI results in Figure 1.

We demonstrate (Figure below, panel f) that preloading EV71 2A to SETD3 cannot be disrupted by subsequent loading of actin. Black curve: SETD3 loaded on SA biosensor was first exposed to EV71 2A and subsequently AP-actin; Red curve: SETD3 loaded biosensor was first exposed to G buffer and subsequently AP-actin.

However, (Figure below, panel g) preloading AP actin with SETD3 had little effect on subsequent binding of EV71 2A. Black curve: SETD3 loaded on biosensor was first exposed to AP-actin and subsequently EV71 2A; Red curve: SETD3 loaded biosensor was first exposed to G buffer and subsequently EV71 2A. These results provide evidence that 2A can disrupt the SETD3-actin interaction via competitive binding. We included these results in the revised figure 3 f-g and in main text. (page 11, line 235-245)

5. If the SETD3-2A interaction is dynamic or transient, does 2A affect methylation activity of SETD3 on actin substrate? If SETD3 is still active on actin in the presence

of A2, it will be more meaningful to measure the change of K_m value from the kinetics study.

The premise of the study seems to suggest that SETD3 and actin form a stable and long-lasting complex. However, what could be the reason(s) for the enzyme to form a stable complex with substrate (or product)?

Responses:

We agree with this reviewer that measuring changes in kinetic parameters is important to address 2A inhibit the MTase activity of SETD3. We determined Michaelis-Menten kinetics of SETD3 catalyzed MTase reaction as previously described⁵.

We determined Michaelis-Menten kinetics for SETD3 catalyzed methyl transfer reaction using a short actin peptide (66-TLKYPIDHGVITNWD-80) as the substrate. The calculated K_m was $8.1 \pm 0.5 \mu\text{M}$ and the turnover number k_{cat} was $18.5 \pm 0.3 \text{h}^{-1}$ (Figure below, Table below). When adding EV71 2A with 2-folds and 5-folds excess over SETD3 in the reaction mixtures, the K_m of the MTase reaction increased to $38.9 \pm 6.8 \mu\text{M}$ and $44.9 \pm 8.8 \mu\text{M}$, respectively; and the k_{cat} dropped to $13.5 \pm 1 \text{h}^{-1}$ and $9.6 \pm 0.8 \text{h}^{-1}$, respectively (Figure below, Table below). These results demonstrate that EV71 2A impaired actin-peptide binding at the SETD3 active site thus inhibited the MTase activity, which supports our hypothesis. We included these results in the revised figure 3e and in main text. (page 11, line 223-234)

	$k_{cat}(\text{h}^{-1})$	$K_m(\mu\text{M})$
SETD3 Alone	18.5 ± 0.3	8.1 ± 0.5
SETD3:2A(1:2)	13.5 ± 1.0	38.9 ± 6.8
SETD3:2A(1:5)	9.6 ± 0.8	44.9 ± 8.8

We apologize for this confusing description that SETD3 and actin form a stable complex. We modified associated description in our revised manuscript. (Page13, line 279-280).

6. 2A Y94 was identified as the key residue for interaction with SETD3; does the

SETD3 F409 mutant (the residue makes interaction with 2A Y94) has the same effect?

Responses:

As we mention above, we replaced our ITC results with BLI results in the revised manuscript. To address this question, we employed the BLI technique to measure the binding of SETD3 with EV71 2A Y94A (Figure below), demonstrating Y94A slightly reduced binding with a K_D of 30.2 nM for Y94A and 22.7nM for WT protein. This is consistent with previous results that mutations in the RSB domain interface have only minor effects on 2A binding (Nat Commun (PMID 36075902)). The F409A and G361R mutant in the RSB domain was proved to have smaller effect on binding affinity to 2A. However, this is inconsistent with the *in vivo* results, making the *in vivo* results confusing and unexplainable. Therefore, we deleted Y94 associated description and discussion in the revised manuscript.

7. Does the cofactor SAM or SAH has any effect SETD3-2A interaction?

Responses:

We measured binding affinity of SETD3 with EV71 2A in the presence of SAH using BLI. The results indicate that SAH has negligible effect on the SETD3-2A interaction (Figure below). We included these results in the revised figure 1h and in main text. (page 6,line120-123)

8. The difference seen between X-ray and cryo-EM models are probably due to the nature of low resolution.

Responses:

Although overall resolution of X-ray structure of SETD3-EV71 2A is low $\sim 3.5 \text{ \AA}$, chain B and F (one copy of SETD3-EV71 2A complex) has clear electron density for model building (Figure below). The cryo-EM structure also has good density for model building. Therefore, we reasoned that these differences seen between X-ray and cryo-EM models were stemmed from different conditions during structure determination, crystal lattice vs EM grids, which may reflect some level of dynamics between SETD3 are 2A. Comparing our structure with SETD3-CVB3 2A structure, we also observe difference between SETD3-2A interactions (Figure below), regardless the SETD3-2A interactions involve largely conserved residues.

9. How did the authors reach the conclusion of 1:1 ratio of SETD3 and A2 in Figure 1C?

Responses:

To determine the stoichiometry of SETD3-EV71 2A complex, we carried out analytical ultracentrifugation (AUC) to calculate molecular mass. From the results (see figure below, panel e), we are confident that the SETD3-EV71 2A complex has 1:1 molar ratio. In detail, the calculated molecular mass of EV71 2A alone is ~17 kDa by AUC; the calculated molecular mass of SETD3 (1-498) is ~54 kDa by AUC, matching their theoretical molecular mass (16 kDa for 2A and 57 kDa for SETD3). The calculated molecular mass of the SETD3-EV71 2A complex is ~66 kDa by AUC, it is very close to its theoretical molecular mass, 73 kDa. We added these new results to the revised manuscript. (page 5, line 98-104).

10. What is the zinc finger of 2A? why is it involved in protein-protein interactions? Any references on the statement?

Responses:

We added description of the zinc finger of 2A and described its involvement in protein-protein interaction in main text, Page3, line62-63. More references are added in the revised paper.

11. The conclusion – SETD3-2A involves more van der Waals and hydrophobic contacts – is not supported by the description.

EV71 2A H71-SETD3 Y288, EV71 2A R115-SETD3 S264, and EV71 2A Q117-SETD3 Q257, EV71 2A K70-SETD3 E296, residues R115 and Q117, SETD3 Q380 and EV71 2A K78 are all charged and polar residues. Q380 salt bridges with K78 is incorrect. The salt bridge only applies to charge-charge interactions, and Q380 is not a charged residue.

Among all the residues being discussed, only four residues F409, L81, Y83 and Y94 are hydrophobic and aromatic residues.

Responses:

We deleted this statement and corrected description of SETD3-EV71 2A interactions in main text, page 9, line180-182.

The functional significance of Y94 cannot be appreciated from the current structure description.

Responses:

As mentioned above, we deleted Y94 associated description and discussion in the revised manuscript.

11. The remarkable conformational rearrangement of R-strand is not clear from Fig. 3D.

Responses:

In the revised Fig. 3, we illustrate detail of conformational rearrangement of R-strand, see figure below. We added more description in main text, page10, line 209-213.

12. Is H71 of 2A located in the same binding site of actin histidine substrate? The details of H71 binding through backbone hydrogen bonds should be shown.

Responses:

Yes, the H71 of 2A located in the same binding site of actin histidine substrate. We prepared a new figure panel (Fig 3d) to illustrate the details of H71 binding through backbone hydrogen bonds. As the figure 3d illustrated (Figure below), the 2A-binding site in SETD3(cyan stick model) partially overlaps with the actin peptide residues 66-72 binding site (red stick model) in SETD3, both forming intermolecular β sheets with SETD3. Two hydrogen bonds are formed between the backbone amide and carbonyl oxygen of SETD3 Y288 with carbonyl and amide of 2A H71(cyan dot line). In the actin-SETD3 complex, the SETD3 Y288 amide forms a hydrogen bond with the carbonyl of actin Y69, and a second backbone hydrogen bond is formed between the SETD3 T286 carbonyl and the amide in the actin peptide I71 (red dot line). We added more description in main text, page10, line215-222.

- 1 Joel, P. B., Fagnant, P. M. & Trybus, K. M. Expression of a nonpolymerizable actin mutant in Sf9 cells. *Biochemistry* **43**, 11554-11559, doi:10.1021/bi048899a (2004).
- 2 Igarashi, H. *et al.* 2A protease is not a prerequisite for poliovirus replication. *Journal of virology* **84**, 5947-5957, doi:10.1128/JVI.02575-09 (2010).
- 3 Yang, X. *et al.* Proteolytic Activities of Enterovirus 2A Do Not Depend on Its Interaction with SETD3. *Viruses* **14**, doi:10.3390/v14071360 (2022).
- 4 Diep, J. *et al.* Enterovirus pathogenesis requires the host methyltransferase SETD3. *Nat Microbiol* **4**, 2523-2537, doi:10.1038/s41564-019-0551-1 (2019).
- 5 Wilkinson, A. W. *et al.* SETD3 is an actin histidine methyltransferase that prevents primary dystocia. *Nature* **565**, 372-376, doi:10.1038/s41586-018-0821-8 (2019).

Reviewer #1 (Remarks to the Author):

The authors have been responsive to earlier suggestions and have significantly improved the manuscript by adding new important experimental data. I view the manuscript to be acceptable for publication.

Reviewer #2 (Remarks to the Author):

I remain unconvinced by the modifications presented in the solution biophysical studies and the derived conclusions, as compared to the previous submission. Given that the previous data are now deemed invalid, I will not delve into detailed comments again.

Reviewer #3 (Remarks to the Author):

The authors have addressed all the previous reviewer comments.